# Immunotherapeutic Approaches for the Treatment of HPV-Associated (Pre-)Cancer of the Cervix, Vulva and Penis

**DOI:** 10.3390/jcm11041101

**Published:** 2022-02-19

**Authors:** Tynisha S. Rafael, Jossie Rotman, Oscar R. Brouwer, Henk G. van der Poel, Constantijne H. Mom, Gemma G. Kenter, Tanja D. de Gruijl, Ekaterina S. Jordanova

**Affiliations:** 1Department of Urology, The Netherlands Cancer Institute, Plesmanlaan 121, 1066 CX Amsterdam, The Netherlands; t.rafael@nki.nl (T.S.R.); o.brouwer@nki.nl (O.R.B.); h.vd.poel@nki.nl (H.G.v.d.P.); 2Department of Obstetrics and Gynecology, Center for Gynecological Oncology Amsterdam (CGOA), Amsterdam UMC, Vrije Universiteit Amsterdam, De Boelelaan 1117, 1081 HV Amsterdam, The Netherlands; j.rotman@amsterdamumc.nl (J.R.); c.mom@amsterdamumc.nl (C.H.M.); g.g.kenter@amsterdamumc.nl (G.G.K.); 3Department of Medical Oncology, Cancer Center Amsterdam, Amsterdam UMC, Vrije Universiteit Amsterdam, De Boelelaan 1117, 1081 HV Amsterdam, The Netherlands; td.degruijl@amsterdamumc.nl

**Keywords:** human papillomavirus, immunotherapy, urogenital, cervical cancer, vulvar cancer, penile cancer

## Abstract

Human papillomavirus (HPV) infection drives tumorigenesis in almost all cervical cancers and a fraction of vulvar and penile cancers. Due to increasing incidence and low vaccination rates, many will still have to face HPV-related morbidity and mortality in the upcoming years. Current treatment options (i.e., surgery and/or chemoradiation) for urogenital (pre-)malignancies can have profound psychosocial and psychosexual effects on patients. Moreover, in the setting of advanced disease, responses to current therapies remain poor and nondurable, highlighting the unmet need for novel therapies that prevent recurrent disease and improve clinical outcome. Immunotherapy can be a useful addition to the current therapeutic strategies in various settings of disease, offering relatively fewer adverse effects and potential improvement in survival. This review discusses immune evasion mechanisms accompanying HPV infection and HPV-related tumorigenesis and summarizes current immunotherapeutic approaches for the treatment of HPV-related (pre-)malignant lesions of the uterine cervix, vulva, and penis.

## 1. Introduction

Human papillomavirus (HPV) is a virus that can be sexually transmitted or nonsexually acquired, primarily through skin-to-skin or skin-to-mucosa contact. Infection with HPV is so common that the lifetime risk in sexually active individuals is around 80–90% [1]. More than 100 HPV genotypes have been identified, of which >40 are able to infect the mucosa [2,3]. The HPV genotypes can be classified as low-risk or high-risk (hrHPV). Infection with low-risk HPV types mainly causes skin lesions such as anogenital warts, whereas persistent infection with hrHPV types increases the risk of dysplasia and cancerous lesions [4]. Moreover, persistent infection with hrHPV types is estimated to be responsible for ~5% of human cancers [5]. It accounts for virtually all cervical cancers [4,6,7], approximately one third of vulvar cancers, and ~50% of all penile cancers [5,8,9]. 

Current treatment options for HPV-related urogenital (pre-)malignancies are often associated with significant treatment-related morbidity and/or toxicity. Early stages of HPV-related cancers of the cervix, vulva, and penis can be surgically treated; however, surgery is often associated with morbidity due to lymphedema, wound infections, and psychosocial and psychosexual problems [10,11,12,13,14]. Indeed, vulvar cancer and penile cancer patients especially experience deleterious effects on sexual function given the large surgical extent in a sexually sensitive area. In addition to surgery, adjuvant chemoradiotherapy may also have short-term and long-term effects. In cervical cancer, long-term effects of chemoradiation on sexual, bladder, and bowel function as well as premature ovarian failure and infertility in premenopausal women have been described. In the setting of advanced HPV-related cancers, patients have limited treatment options that offer poor and nondurable clinical responses [10,15,16]. The occurrence of treatment-related morbidities and poor survival in the advanced or metastasized setting highlight the need for novel, well-tolerated, and most importantly, durable therapies that prevent recurrent disease and improve clinical outcome. 

Immunotherapy can be a useful addition to the current therapeutic strategies in various settings of disease, offering relatively fewer adverse effects and potential improvement in survival. Here, we discuss several immune evasion mechanisms accompanying HPV infection and HPV-related tumorigenesis and summarize current immunotherapeutic approaches for the treatment of HPV-related (pre-)malignant lesions of the uterine cervix, vulva, and penis.

## 2. HPV Infection, Prevention, and Related Urogenital (Pre-)Malignancies

### 2.1. HPV Infection

Epithelial cells in the undifferentiated basal layer can be infected by hrHPV types as a result of micro-abrasions in the cutaneous or mucosal epithelium [2,17]. The infected basal keratinocytes form a reservoir of infection, and as these cells divide, viral replication moves towards the suprabasal layers (the midzone and superficial zone) of the epithelium. HPV virions are released from the cornified keratinocytes and shed viral particles that can then initiate a new infection. 

### 2.2. Prophylactic Vaccines Play an Important Role in Preventing Initial HPV Infection

Three prophylactic vaccines for prevention of HPV infection are currently available: bivalent (targeting HPV 16 and 18), quadrivalent (targeting HPV 6, 11, 16, and 18), and nonavalent (targeting HPV 6, 11, 16, 18, 31, 33, 45, 52, and 58). All vaccines are composed of L1 virus-like particles (VLPs) that are specific for the HPV types being targeted [18]. These vaccines are highly immunogenic and induce specific HPV-neutralizing antibodies. Prophylactic HPV vaccination of both girls and boys protects against initial infection. Vaccination in combination with cervical screening programs can eliminate the risk for cervical cancer, as well as other HPV-related malignancies [19]. A recent observational study showed a corresponding risk reduction of 97% for precursor lesions of the cervix and an 87% reduction for cervical cancer for those vaccinated at age 12–13 years [20]. This highlights the importance of implementing HPV immunization programs worldwide. Unfortunately, vaccination coverage in most countries is still suboptimal and varies both between and within countries [21,22,23]. Another obstacle is that in many countries boys and men are vaccinated at low rates. In 2019, only 4% of boys had received the full course of the vaccine worldwide [22]. At the moment, vaccination of boys is recommended to reduce HPV prevalence and has been included in nationwide school-based vaccination programs across the globe [23]. Models predict that vaccination of boys, in addition to girls, can eliminate HPV 6, 11, 16, and 18, if 80% coverage is achieved [24]. However, considering existing societal barriers against vaccination and the increasing incidence rates for HPV-related cancers, many men and especially women will still have to face HPV-related morbidity and mortality in the upcoming years. 

### 2.3. HPV-Related (Pre-)Cancer of the Cervix, Vulva, and Penis

#### 2.3.1. Cervical Cancer and Precursor Lesions

Cervical cancer is the fourth most frequently diagnosed cancer in women, with an estimated 604,000 new cases and 342,000 deaths worldwide in 2020 [25]. The two main histological subtypes, squamous cell carcinoma (SCC) and adenocarcinoma (AC), account for 65% and 30% of the cases, respectively [26]. HPV 16 is often associated with SCC, and HPV 18 with AC [27]. Precancerous cervical intraepithelial neoplasia (CIN) lesions are categorized as low-grade (CIN1) and high-grade (CIN2 and 3). CIN develops in the transformation zone, which is an area of metaplastic tissue between the squamous epithelium of the vagina and ectocervix and the glandular tissue of the endocervical canal. CIN2 and CIN3 are often treated by excision or ablation to remove the affected transformation zone. 

The standard of treatment for early stage cervical cancer is radical hysterectomy with pelvic lymphadenectomy or primary radiation therapy [28]. Adjuvant radiotherapy, with or without chemotherapy, is considered when patients present risk factors for disease recurrence (e.g., positive surgical margins, lymph node metastasis, and/or parametrial involvement) [29,30,31,32]. In locally advanced cervical cancer, treatment consists of definitive chemoradiotherapy [33]. In the setting of metastasized or recurrent disease, patients are treated with chemotherapy and bevacizumab (targeting vascular endothelial growth factor (VEGF)) [34]. 

#### 2.3.2. Vulvar Cancer and Precursor Lesions

Vulvar cancer is a rare type of cancer. Its incidence has been increasing annually and currently stands at approximately 45,000 new cases and 17,000 deaths worldwide in 2020 [25]. Approximately 90% of all vulvar cancers are vulvar squamous cell carcinomas (VSCC) [35]. Two main oncogenic pathways have been identified in the pathogenesis of VSCC: (1) an HPV-dependent pathway that starts with infection with hrHPV and (2) an HPV-independent pathway that is associated with chronic inflammation (e.g., lichen sclerosus) with either wildtype p53 or mutated p53 [36,37,38]. Patients with HPV-related vulvar cancer have a more favorable prognosis, while patients with HPV-negative tumors carrying p53 mutations have the worst prognosis [37]. Nevertheless, all patients receive the same treatment regardless of HPV or p53 status. Vulvar intraepithelial neoplasia (VIN) is considered a precursor of VSCC and can be divided into low-grade squamous intraepithelial lesions (LSIL), vulvar high-grade squamous intraepithelial lesions (vHSIL), or differentiated VIN (dVIN). dVIN is the precursor of hrHPV-negative VSCC and has been associated with worse disease-free survival compared to HSIL-associated vulvar cancer [39]. vHSIL is the precursor of hrHPV-positive VSCC and was formerly referred to as vulvar intraepithelial neoplasia of the usual type (uVIN).

Management options for precursor lesions are surgical excision for dVIN and surgical excision, laser evaporation, or topical treatment for vHSIL. The standard treatment for VSCC consists of radical excision, if feasible. The presence of inguinofemoral lymph node (LN) metastases is the most significant prognostic factor for survival [40]. For this reason, sentinel LN procedure, lymphadenectomy, or LN debulking is performed as well, except for women with stage IA disease (lesions ≤ 2 cm and stromal invasion ≤ 1 mm). Chemotherapy and radiation are applied as adjuvant or primary treatment for disease that cannot be surgically resected. 

#### 2.3.3. Penile Cancer and Precursor Lesions

Penile cancer is a rare malignancy with an estimated 36,000 new cases and 13,000 deaths worldwide in 2020 [25]. More than 95% of penile tumors are penile squamous cell carcinoma (PSCC) [41]. Similar to VSCC, two molecular pathways of etiology have been described: HPV-dependent and HPV-independent carcinogenesis [42,43]. hrHPV+ patients tend to have better disease-specific survival compared to hrHPV− PSCC patients [44,45,46,47,48]. Penile intraepithelial neoplasia (PeIN) is considered to be the precursor lesion for PSCC and is classified as being either undifferentiated (HPV-related) or differentiated (non-HPV-related) [10,41,49]. HPV-related lesions are usually found on the penile glans and/or foreskin with basaloid/warty features [10,41]. Non-HPV-related lesions can be characterized by atrophic and hyperplastic epithelium and are typically associated with underlying lichen sclerosus [42]. Treatment options for precursor lesions include therapeutic circumcision, topical therapies, laser ablation, and/or local excision or total glans resurfacing. The location, extent, persistence, and recurrence rate of the lesions determine the type of treatment [10]. 

Once progressed to invasive disease, the excision technique used for management of disease depends on the stage at diagnosis. Organ-sparing surgical techniques have been developed for cT0–2 disease to ensure maximal organ preservation [50]. On the other hand, large tumors (cT3–4) may require more aggressive options, such as partial or total penectomy [10]. Moreover, patients presenting with inguinal and/or pelvic lymph node metastasis are generally treated with lymphadenectomy, which is associated with considerable morbidity. Unresectable, locally advanced, or metastatic PSCC requires multimodal approaches [10].

## 3. Immune Evasion Strategies of HPV and HPV-Related Cancer and Precursor Lesions of the Cervix, Vulva, and Penis

HPV-related urogenital tumors are associated with improved survival, most likely due to their increased immunogenicity that is related to antiviral immune responses. This in turn requires immune evasion strategies in order for the tumor to grow and invade. Several intracellular evasion strategies of HPV, which are mediated by altered gene expression and disturbed protein functions, and extracellular strategies, which are mediated by interfering with immune cell networks, have been described [51]. Clearly, it is important to identify all these possible immune evasion mechanisms in order to achieve optimal efficacy of immunotherapy. The most important ones established so far are summarized below (Figure 1).

### 3.1. Intracellular Immune Evasion Strategies of HPV

In the early phases of infection, HPV infection primarily involves the basal layer of the stratified epithelium, where the virus has the capability of maintaining low abundance of viral proteins, resulting in low-level immunogenicity [52]. Additionally, since HPV virions are only released at the epithelium surface without inducing cytolysis and inflammation [53,54,55], there is a limited to almost absent release of danger signals. This impairs the activation of pathogenic recognition receptors (PRR), such as toll-like receptors (TLRs), interferon-gamma-inducible protein 16 (IFI16), cyclic GMP-AMP synthetase (cGAS), and retinoic-acid-inducible gene I (RIG-I), and ensures the suboptimal activation of the local immune response. However, during disease progression, the expression of HPV oncoproteins E6 and E7 starts to increase [51,56]. To evade immune recognition and establish persistence of infection, HPV E6 and E7 are able to modulate host gene expression via (1) the activation of the transcription factors STAT3 and NFκB, and (2) the antagonism of several pathogen sensors (such as TLRs, IFI16, cGAS, and RIG-I) [51,56,57,58]. In this manner, important signaling pathways (such as cGAS-STING, TLR9, and NFκB) are dysregulated at different stages [51,59,60,61]. HPV E7 is known to epigenetically repress the expression of the adhesion molecule E-cadherin, which is necessary for adhesion of Langerhans cells (LC) to keratinocytes [62]. Next to downregulated antiviral gene expression and disturbed cell-to-cell interactions, HPV-transformed cells display impaired production of major histocompatibility complex (MHC) class I and II components of the antigen-processing machinery (APM), resulting in less immune recognition due to reduced presentation of HPV epitopes [54]. 

### 3.2. Extracellular Immune Evasion Strategies of HPV-Related (Pre-)Cancers

In addition to intracellular immune evasion strategies, HPV employs several extracellular strategies that prevent a robust immune response through perturbation of cellular immune networks that are vital for the clearance of infection [51]. All these strategies enable the virus to persist for a long time, which increases the risk of (pre-)malignancy. Furthermore, HPV-driven tumors have their own tissue-specific immune escape mechanisms. Understanding these mechanisms involved in immune escape of HPV-tumors may help us find clinical opportunities for immune interventions at the various steps of disease development. 

#### 3.2.1. Antigen-Presenting Cells 

Antigen-presenting cells (APCs) initiate the cellular immune response and are the connection between the innate and adaptive immune system. They capture, process, and present viral antigens on MHC molecules. In the upper epithelial layers of the epidermis, LCs (members of the dendritic cell (DC)/macrophage family) detect viral structures through their TLRs. Upon recognition and uptake of antigens, they migrate to tumor-draining lymph nodes (TDLNs), where they can cross-talk with lymph-node-resident DC subsets and mediate the priming and activation of effector T-cell responses against the HPV-derived and cancer cell-related antigens. However, several immune evasion strategies have been linked to defective or inhibited T-cell priming in the TDLNs. For example, different studies have shown that HPV virus-like particles are not able to properly activate LCs and prime T-cells [63,64,65]. Moreover, HPV E7 disrupts LC retention in epithelial tissue through the downregulation of E-cadherin [62,66]. HPV E6 and E7 in keratinocytes can also downregulate the release of the chemokine CCL20, thereby preventing migration of immature LCs into the epidermis [67]. Indeed, in cervical lesions, active HPV infection and/or expression of E6 and E7 was associated with decreased frequencies of intraepithelial LCs [67,68,69]. In addition, in vHSIL, decreased numbers of CD1a+ migratory DCs and Langerin+ DCs were found compared to healthy tissue [70].

Not only is the frequency of DCs of significance for an effective HPV-specific immune response, but so to is their activation status. In low-grade CIN lesions, reduced expression of MHC-II and Langerin on LCs is observed compared to normal cervical tissue [69]. The expression of the T-cell-activating costimulatory molecules CD80 and CD86 on CD11c+ DCs decreases with increasing CIN grade, resulting in a poor antigen-presenting environment [71]. Cervical and vulvar cancer cells can inhibit the maturation and function of DCs through the secretion of factors such as IL-6, prostaglandin E2 (PGE2), receptor activator of NF-KB ligand (RANKL), and indoleamine 2,3-dioxygenase (IDO) [72,73,74,75]. Altogether, HPV-infections seem to be associated with less activation and migration of DCs, which may lead to defective or inhibited T-cell priming in the TDLNs and thereby the absence of an effective T-cell response.

#### 3.2.2. T-Helper Cell and Cytotoxic T-Cell Responses

Robust HPV-specific CD4+ T-helper cell and CD8+ T-cell responses are necessary for effective elimination of HPV-infected cells. After interaction with mature APCs, naïve CD8+ T cells can differentiate into cytotoxic CD8+ T cells (CTLs) and then migrate to the tumor bed and recognize cancer cells through the interaction between their T cell receptor (TCR) and MHC-I-bound antigens. However, CTL recognition is prevented by downregulation of MHC-I molecules on HPV-infected keratinocytes [76,77,78,79]. CTLs can also be present but functionally inactive within the tumor microenvironment, as demonstrated by their limited and/or absent production of IL2, granzyme B (GrB), and IFNγ [80,81]. Furthermore, regression rates of CIN lesions are strongly correlated with the presence of intraepithelial GrB+ CTLs [82]. In vulvar and penile cancer, the number of GrB+ intraepithelial lymphocytes was related to better outcomes [83,84]. 

Proper CD4+ T-helper (Th) cell responses are also needed for clearance of HPV and for durable T-cell memory. Th1-type responses (characterized by high levels of IFN-γ and IL-2) are required for an effective and durable anti-HPV immune response [85,86]. Persistent infection is known to promote Th2 responses, which are characterized by high levels of IL-6, IL-8, and IL-10 [87]. In CIN, a shift from Th1 to Th2 response has been associated with progression of lesions [87,88,89,90]. This has not yet been established in vHSIL and PeIN. 

#### 3.2.3. Regulatory T Cells

Foxp3+ regulatory T cells (Tregs) are key mediators of immune suppression in the tumor microenvironment. Tregs exert their immunosuppressive function via multiple mechanisms: (1) the secretion of the cytokines IL-10, TGF-β, IL-35 and through direct cell-to-cell contact via membrane-bound TGF-β and cytotoxic T-lymphocyte-associated protein 4 (CTLA-4) [91,92]. The mechanisms by which Tregs are recruited and activated in HPV-infected tumors are not fully understood. It is believed that Tregs can possibly originate from both thymus-generated natural Tregs and peripheral inducible Tregs [93]. The latter are generated during priming by activated DCs in an anti-inflammatory milieu with high expression levels of cytokines such as TGF-β and IL-10. These host-derived cytokines are produced by suppressive immune subsets and tumor cells in the local microenvironment and help activate and expand Tregs [91,94,95]. Moreover, the activation and expansion of Tregs may also be induced by their interaction with M2 macrophages [96,97,98]. In CIN, Tregs are recruited and activated at higher frequencies in the blood of patients with persistent HPV infection compared with patients that had cleared infection or had detectable E6-specific CD4+ T-cell responses [99,100]. Patients with uVIN more often have recurrences when high numbers of intraepithelial Tregs are present [96]. In patients with cervical cancer and penile cancer, increased frequencies of Tregs were associated with poor clinical outcome [77,101,102]. Remarkably, high numbers of Tregs in cervical SCC were associated with poor outcome [77,101], whereas in the AC tumor microenvironment, an opposite association was observed [103]. The latter may be due to co-infiltration of Tregs with conventional effector T cells, which overall is less in AC than in SCC [104] 

#### 3.2.4. Myeloid Cells

HPV-transformed cells are able to recruit and alter the phenotype and functionality of myeloid cells through the release of various soluble factors. Macrophages are a heterogeneous population of monocyte- or tissue-resident precursor-derived myeloid phagocytes with extraordinary plasticity. Macrophages are typically categorized into the pro-inflammatory M1 phenotype (CD68 + CD80+) and an immunosuppressive M2 (CD68 + CD163+) phenotype based on an in vitro polarization system [105]. However, macrophages in vivo have more complex phenotypes in the tumor microenvironment beyond this simple categorization, which merely represents two ends of a continuous spectrum [105,106]. In cervical cancer tissue, higher numbers of intraepithelial M2 tumor-associated macrophages are present compared to nontumorous cervical tissue [107]. In cervical cancer, increased numbers of M2 macrophages were associated with disease progression and worse clinical outcome [108,109]. On the other hand, the presence of M1 macrophages (CD14 + CD33–-CD163–) in the microenvironment was associated with an influx of intraepithelial T cells and a good prognosis [110]. Interestingly, CD163+ M2-like macrophages appeared to be recruited to tumor-involved cervical TDLN, where they formed an immune-suppressive cordon around the tumor nests and their numbers were perfectly correlated to Treg rates, suggesting co-regulation or possibly macrophage-mediated Treg expansion [98,111]. However, the finding that Treg recruitment to TDLN appeared to precede both tumor invasion and subsequent M2 recruitment rather suggested a driving role for Tregs in tumor progression and an associated M2 macrophage influx [111]. No association was found between CD14+, CD68+, and CD163+ myeloid densities and survival in patients with PSCC [112]. 

In addition to macrophages, monocytic or polymorphonuclear myeloid-derived suppressor cells (MDSC) derived from early myeloid precursors can also contribute to T cell suppression when conditioned by tumor-derived suppressive cytokines, such as IL-6, VEGF, and TGF-β. Indeed, both MDSC subsets were found at increased frequencies in cervical TDLN, and monocytic MDSC in vulvar TDLN, upon metastatic involvement [98,113]. 

#### 3.2.5. T-Cell Activation and Inhibition

T cells are activated after presentation of antigens on MHC I/II molecules by APCs to T-cell receptors (TCR). Antigen recognition alone is not enough for the full induction and activation of T cells; co-stimulation is also needed and results, amongst others, from the interaction of CD28 receptors on T cells and CD80/CD86 on APCs. T-cell anergy and apoptosis ensue in the absence of co-stimulation [114]. To prevent uncontrolled T-cell activation, immune checkpoint molecules bind to their respective ligands [115]. The tumor microenvironment is enriched with immunosuppressive factors, inducing checkpoint receptor expression on T cells, which promotes T cells with an exhausted/dysfunctional phenotype, ultimately resulting in immune escape [116].

One of the most studied immune checkpoint pathways is that of programmed death-1 receptor (PD-1) and PD-ligand-1 (PD-L1) and PD-ligand-2 (PD-L2). PD-1 is a co-inhibitory receptor on activated T cells that has an immunoregulatory function in normal situations. Interaction of PD-1 on T cells with PD-L1 and PD-L2 on several types of myeloid cells and tumor cells triggers a cascade of downstream signals, resulting in the inhibition of signaling through the TCR complex and CD28 [117]. Inhibition of TCR signal transduction and CD80 co-stimulation results in T-cell functional exhaustion and anergy [118,119]. In HPV-related cervical, vulvar, and penile (pre-)cancer, multiple immunohistochemistry (IHC) studies have shown that the expression of PD-1 on T-cells and PD-L1 on myeloid cells and tumor cells is common [71,84,120,121,122,123,124,125,126,127,128,129,130,131,132]. The exact underlying mechanism for this upregulation remains to be elucidated [133]. Moreover, in SCC of both the cervix and penis, patients with diffuse PD-L1 expression had a worse survival compared with patients with tumor–stroma margin PD-L1 expression [84,126,130], highlighting the importance of expression patterns of PD-L1. The observed marginal expression of PD-L1 may be related to the vicinity of IFNγ-releasing effector T cells, which fits with the apparent association with improved survival [126,134,135]. 

Another well-studied immune checkpoint is CTLA-4. This is a receptor that is expressed on activated T-cells and especially on Tregs. CTLA-4 competes with CD28 receptor for binding to CD80 and CD86 on APC, and binding of antagonistic antibodies may lead to Treg depletion based on antibody-dependent cytotoxicity mediated by (non-)classical monocytes [136]. The expression of additional immune checkpoints in various cancer types has also been described, such as T-cell immunoglobulin and mucin-domain-containing-3 (TIM-3), lymphocyte activation gene-3 (LAG-3), and T-cell immunoglobulin and ITIM domain (TIGIT) [116]. TIM-3 is expressed on fully differentiated Th1 cells and activated by binding to its ligand galectin-9 [137]. This leads to termination of Th1-driven immunity and an increased Tregs suppressive activity [137,138,139]. In cervical cancer, various immune checkpoints were often co-expressed on effector T cells and elevated upon metastatic involvement in TDLN [140], observations consistent with the acquisition of an exhausted phenotype. In vulvar TDLN, beside PD-1 upregulation, in particular high expression levels on T helper cells of CTLA-4 were noticeable upon tumor invasion, which, together with high Treg frequencies and decreased rates of effector T cells, suggested that combined CTLA-4 and PD-1 blockade might help block metastatic spread of this tumor type [113]. In penile cancer, higher frequencies of CD4+ and CD8+ T cells that express CTLA-4 and PD-1 were found in LN containing metastases, indicative of an immunosuppressed microenvironment (Rafael et al., manuscript in preparation).

## 4. Restoring Immune Cell Function in the HPV-Related Tumor Microenvironment 

Immune cells present in the microenvironment of HPV-related (pre-)cancers are often inhibited in their antitumor function. Given the current knowledge on evasion strategies of HPV and HPV-driven tumors, different strategies at different stages of disease development are being employed to restore immune cell function in HPV-infected microenvironments (Figure 2). This section summarizes current results based on studies investigating immunotherapeutic strategies for HPV-related (pre-)cancer of the uterine cervix, vulva, and/or penis.

### 4.1. Toll-like Receptor Agonists

There are several TLR agonists that are being tested for their ability to boost anticancer immune response in HPV-related (pre-)cancers. Imiquimod, an TLR7 agonist, can stimulate local production of pro-inflammatory cytokines such as IFN-γ, TNF-α, and IL-12, which, in turn, can induce type 1 and cytolytic T-cell responses [141,142]. The induction of these responses can initiate immune clearance of HPV-infected cells. Patients with HPV-related intraepithelial neoplasia are increasingly being treated with topical imiquimod treatment instead of excision or ablation. In patients with vHSIL lesions, imiquimod is already used as an initial topical therapy [143]. So far, randomized controlled trials have revealed complete response in 46–58% of patients with vHSIL [143,144]. This therapy also seems to be effective in treating CIN and PeIN lesions, with complete response rates of 67–75% for CIN2–3 and 62% for PeIN [145,146]. These encouraging effects, however, should be interpreted with caution since the available evidence is based on studies lacking uniformly defined endpoints and/or controlled trial data. Moreover, although topical imiquimod is a good alternative for otherwise mutilating treatments, it is of interest to identify determinants of response and resistance to therapy. Several studies have shown that responsiveness to imiquimod was associated with the presence of a pre-existing pro-inflammatory immune microenvironment [147,148,149]. In both vHSIL and CIN, the immune microenvironment of complete responders prior to imiquimod comprised a coordinated infiltrate of type 1 (Tbet+) T-cells as well as pro-inflammatory M1 macrophages [147,148]. The infiltration of CD4+ and CD8+ T-cells was also further amplified after topical imiquimod application in the complete responder group [147,150]. Nonresponsiveness to imiquimod was related with increased proportions of Tregs, limiting the action and development of any HPV T-cell immunity [150]. No studies determining such effects in PeIN have been performed yet. These findings suggest that imiquimod can amplify T-cell responses but is mostly effective in settings where the lesions comprise a pre-existing pro-inflammatory immune microenvironment. In order to break down barriers existing in lesions with local immune suppression, additional immunotherapeutic strategies (e.g., therapeutic HPV vaccination) should be employed to help tip the balance of immune equilibrium in favor of the host effector response. Indeed, in vitro stimulation of tumor-containing TDLN single-cell suspensions from patients with either cervical or vulvar cancer with different TLR agonists failed to overcome apparent T-cell anergy but instead increased the release of immune-suppressive cytokines, such as IL-6 and IL-10 [113,140]. This is in accordance with the need for additional therapeutic immune modulation.

### 4.2. Therapeutic HPV Vaccination 

Rather than generating HPV-neutralizing antibodies, therapeutic HPV vaccines aim to restore or prime cell-mediated immunity through the induction of HPV-specific T-cell responses [151]. Given that E6 and E7 oncoproteins are constitutively active in HPV-transformed cells (and needed to maintain the transformed phenotype) and absent in healthy cells, they are the ideal targets for a therapeutic vaccine [152]. Several therapeutic HPV vaccines targeting E6 and E7 in HPV-related (pre-)cancers have been investigated, including genetic vaccines (e.g., DNA/RNA/virus/bacterial), protein-based, peptide-based, or dendritic-cell-based vaccines [151]. 

DNA vaccination forms an attractive approach for the induction of cellular immune responses, as these vaccines are very stable and tolerable for all patient populations and easy to produce and relatively cheap [153]. The disadvantages of DNA vaccines are their low transfection efficiency and restricted immunogenicity [154]. A few strategies facilitating antigen delivery, processing, and presentation have been widely adopted to help increase the immunogenicity of HPV vaccines [151,155,156]. Clinical trials have reported on the enhanced immunization by electroporation-delivered DNA vaccine. Electroporation at the injection site can increase cell membrane permeability and enhanced nucleic acid uptake and subsequent immunogenicity [157]. In a phase I trial in patients with CIN3, vaccination with GX-1183 by electroporation elicited a significant E6/E7-specific IFN-γ-producing T-cell response in all nine patients, and 7/9 (78%) of the patients had complete regression of their lesion and clearance of HPV DNA [156]. A Phase II study in a larger population found that 52% (33/64) of the patients had histopathologic regression of CIN3 [158]. Another electroporation-delivered vaccine, VGX-3100, elicited robust adaptive immune responses and provided complete histological regression for 49.5% (53/107) of the CIN2/3 patients [159]. Ex vivo immunological analyses demonstrated that the magnitude of the T-cell response against E6 was associated with clinical outcome. Currently, VGX-3100 is in a Phase III clinical trial (NCT03185013) and is being investigated in 201 patients with confirmed HPV-16/18-positive CIN2/3. Another vaccination strategy to enhance immunization is by DNA tattooing. This strategy showed an increased vaccine-specific T-cell response in comparison with classical intramuscular DNA vaccination in non-human primates [160]. A recent phase I/II clinical trial performed by our group used DNA tattooing technique to deliver a genetically enhanced vaccine targeting E6 and E7 in patients with uVIN [153]. The vaccine was found to be well tolerated, and importantly, 6/14 patients showed an objective clinical response (43%; 14% CR, 29% PR). Systemic HPV-specific T-cell responses were observed in five out of the six responders. Moreover, in a similar patient population, lesion clearance was related to the magnitude of the HPV-specific response ex vivo after vaccination with HPV 16 synthetic long peptide [161,162,163]. 

In one of these studies, remarkable response rates were observed in patients with VIN upon vaccination with a synthetic long peptide vaccine encompassing the HPV 16 E6 and E7 oncoproteins [162]. At 12 months of follow-up, 15 out of 19 patients (78%) had a clinical response, with a complete response observed in 9 out of 19 patients (47%). These responses were accompanied by the induction of type-1 T-cell responses against E6 and E7. Results from a phase II study investigating the same HPV 16 synthetic long peptide vaccine in advanced or recurrent HPV-16-induced gynecological carcinomas were disappointing; monotherapy with this vaccine showed only weak T-cell responses and no clinical benefit [164]. The reason for this failure in late-stage cancer patients is likely due to high immune suppression in both the tumor and the associated lymph nodes. Therapeutic HPV vaccination in combination with other treatment modalities (e.g., checkpoint inhibitors) is therefore needed in order to overcome immune suppression and establish effective anti-immune responses in HPV-related cancers. Indeed, in a phase II study (NCT02426892), the ISA101 synthetic long peptide vaccine in combination with nivolumab (anti-PD-1) resulted in a response rate of 33% (8/24) and median survival of 17.5 months in patients with incurable HPV-16-positive malignant neoplasms (22 oropharyngeal, one cervical, and one anal) [165]. Moreover, several ongoing basket trials are investigating HPV vaccines in combination with other immunotherapy agents in locally advanced or metastatic HPV-positive malignancies (NCT04432597, NCT03439085, NCT04287868).

Overall, for both CIN and uVIN patients, clinical efficacy of therapeutic vaccination was associated with the strength of the vaccine-induced immune response [153,156,159,161,162,166]. Although these results are encouraging, no therapeutic vaccines have been approved for clinical use in patients with HPV-related lesions of the cervix and vulva yet. In the case of PeIN, the usefulness of therapeutic vaccines has yet to be investigated.

### 4.3. Immune Checkpoint Inhibitors in HPV-Related Cancer of the Cervix, Vulva, and Penis 

#### 4.3.1. Immune Checkpoint Inhibitors and Cervical Cancer—Results

Immune checkpoint therapy blocking PD-1, PD-L1, and/or CTLA-4 has proven to be successful in several cancer types, usually characterized by high mutation burdens and a relatively dense T-cell infiltrate, such as melanoma or non-small-cell lung cancer (NSCLC) [167,168]. In HPV-related cancers, the largest studies on immune checkpoint inhibitors (ICI) occurred in head and neck squamous cell carcinoma (HNSCC) [169]. Altogether, these studies indicated activity of immune checkpoint inhibitors, which led anti-PD-1 immune checkpoint blockade to become the standard first- and second-line treatment for recurrent and metastatic HNSCC [170,171,172,173]. In other HPV-related cancers such as cervical cancer, the current evidence of efficacy for ICI is mostly supported by single-arm studies and two randomized controlled trials. Based on the results from the KEYNOTE-826 (NCT03635567) randomized Phase III trial, pembrolizumab (anti-PD-1) has been approved by the US Food and Drug Administration (FDA) as of October 2021 for use as first-line treatment in combination with chemotherapy ± bevacizumab for patients with persistent, recurrent, or metastatic cervical cancer whose tumors express PD-L1 (combined positive score (CPS) ≥ 1) [174]. In the second-line setting, based on the results from the phase II KEYNOTE-158 (NCT02628067), pembrolizumab gained FDA approval as treatment for patients with PD-L1-positive cervical cancer (CPS of ≥1) in 2018 [175]. Table 1 summarizes the (preliminary) results of clinical trials investigating ICI in cervical cancer. 

#### 4.3.2. First-Line Treatment for Recurrent or Metastatic Cervical Cancer

To our knowledge, (preliminary) results from two phase III trials evaluating checkpoint inhibitors in comparison with current standard of care first-line therapy have been published. 

In the phase III, randomized, double-blind, placebo-controlled KEYNOTE-826 trial (NCT03635567), 617 patients with persistent, recurrent, or metastatic cervical cancer were enrolled irrespective of PD-L1 expression status and randomly assigned to receive pembrolizumab (anti-PD-1) 200 mg or placebo every 3 weeks for up to 35 cycles plus platinum-based chemotherapy with or without bevacizumab [174]. In 548 patients with a PD-L1 CPS ≥ 1, median progression-free survival was 10.4 months in the pembrolizumab group and 8.2 months in the placebo group (HR: 0.62; 95% confidence interval (CI), 0.50 to 0.77; *p* < 0.001). Moreover, the benefit of pembrolizumab was shown to increase with increasing PD-L1 expression. Based on the KEYNOTE-826 trial, the FDA has recently approved pembrolizumab in combination with chemotherapy, with or without bevacizumab, for the first-line treatment of patients with persistent, recurrent, or metastatic cervical cancer whose tumors express PD-L1 (CPS ≥ 1).

The randomized, phase III GOG 3016/ENGOT-cx9 trial (NCT03257267) is evaluating the role of monotherapy with cemiplimab (anti-PD-1) in patients with recurrent or metastatic cervical cancer that progressed after platinum-based chemotherapy [176]. In this trial, patients were treated with either cemiplimab 350 mg every 3 weeks (*n* = 304) or investigator’s choice of chemotherapy for up to 96 weeks (*n* = 304). Interim analysis showed an improved overall survival in cemiplimab group in comparison with chemotherapy group (median survival, 12.0 vs. 8.5 months; *p* < 0.001). Furthermore, cemiplimab activity was observed regardless of PD-L1 status or histology. 

#### 4.3.3. Trials with PD-1/PD-L1 or CTLA-4 Inhibitor Therapy in Cervical Cancer—Second-Line

In the phase Ib KEYNOTE-028 study (NCT02054806), the safety and efficacy of pembrolizumab (anti-PD-1) therapy was investigated in patients with PD-L1+ advanced solid cancers, including cervical SCC (*n* = 24) [177]. Patients with previously treated locally advanced or metastatic cervical cancer received pembrolizumab 10 mg/kg every 2 weeks for up to 24 months. Overall response rate was 17% (7/24); four patients (17%) achieved a confirmed partial response (PR) and three patients (13%) had stable disease (SD). All patients discontinued treatment during this study, because of physician’s decision, disease progression, or adverse events (AEs). Grade 3 treatment-related AEs were observed in 21% of the patients. Furthermore, based on the subsequent phase II KEYNOTE-158 basket study (NCT02628067) including a cohort of patients with advanced cervical cancer, pembrolizumab gained FDA approval as a second-line treatment for patients with PD-L1-positive cervical cancer (CPS of ≥1) in 2018 [175,178]. Patients with recurrent and/or metastatic cervical carcinomas after ≥1 prior chemotherapy regimens (*n* = 98) were recruited regardless of PD-L1 status and received pembrolizumab 200 mg every three weeks for up to 2 years. The objective response rate was 12% (12/98; 3 CR, 9 PR). All responses were seen in patients with PD-L1+ tumors (CPS of ≥1 based on 22C3 assay). Treatment-related grade 3 or 4 AEs were seen in 12% of patients. 

In the phase I/II CheckMate 358 trial (NCT02488759), an encouraging response rate of 26.3% (4/19; 3 CR, 1 PR) was observed after nivolumab (anti-PD-1) administration in patients with previously treated recurrent or metastatic cervical cancer [179]. These patients received nivolumab 240 mg every 2 weeks for up to 2 years. Remarkably, responses to nivolumab were not related to PD-L1 status or previous treatments. Treatment-related grade 3 or 4 AEs were seen in 21% of patients. In another phase II trial (NCT02257528), 26 patients with persistent/recurrent cervical cancer previously treated with platinum-based chemotherapy received nivolumab 3 mg/kg IV every 2 weeks until disease progression or intolerable toxicity [180]. Objective response rate was only 4%; one patient obtained a confirmed PR (4%) and nine patients had SD (36%). No significant correlation was found between PD-L1 expression and objective tumor response to nivolumab. Treatment-related grade 3 or 4 AEs were seen in eight patients (32%).

Another promising anti-PD-1 agent is balstilimab. Results from a recent phase II trial (NCT03104699) investigating balstilimab showed encouraging responses and durable clinical activity in patients with recurrent and/or metastatic cervical cancer [181]. Patients (*n* = 161) received balstilimab 3 mg/kg once every two weeks, for up to 24 months. The objective response was 15% and included 5 patients with a complete response and 16 with a partial response. Responses were durable with a median duration of response of 15.4 months. Tumor responses were observed irrespective of squamous cell histology or PD-L1 status. Treatment-related AEs (≥grade 3) were seen in 19 patients (12%).

In a phase I/II trial investigating the effects of ipilimumab (anti-CTLA-4) monotherapy in patients with metastatic or recurrent cervical cancer (*n* = 42), there was an overall response rate of only 2.9% (1/42; 1 PR) [182]. In another phase I study, the GOG 9929 trial (NCT01711515), the safety and efficacy of adjuvant ipilimumab following chemoradiation (CRT) therapy in newly diagnosed node-positive cervical cancer patients was investigated. This combination was well tolerated with possible clinical activity [183]. Treatment-related grade 3 AEs were seen in 9.5% of patients. Of note, inducible T-cell costimulator (ICOS) and PD-1 expression increased on T-cell subsets following CRT and was sustained with sequential anti-CTLA-4 immunotherapy [184]. 

In a phase I trial (NCT02383212), cemiplimab (anti-PD-1) was investigated as monotherapy or in combination with hypofractionated radiation therapy (hfRT) in patients with recurrent or metastatic cervical cancer [185]. Patients in the monotherapy cohort were treated with cemiplimab 3 mg/kg every 2 weeks for up to 48 weeks (*n* = 10), and patients in the combination cohort received additional hfRT in week 2 (*n* = 10). In each cohort, one patient experienced a partial response and both patients had squamous histology. Regarding PD-L1 status, correlative analysis of PD-1 expression from study patients was not performed due to insufficient tumor material. Treatment-related AEs (≥grade 3) occurred in 1/10 monotherapy patients and in 3/10 patients in the combination cohort. 

#### 4.3.4. Combination of Checkpoint Inhibitors in Cervical Cancer

Single-agent immune checkpoint inhibitor administration in patients with cervical cancer has provided encouraging, but modest clinical efficacy with often short-lived benefit. The failure of single-agent monotherapy may be attributable to the fact that there are other mechanisms of immune evasion involved. In order to improve responses to immunotherapy, combined approaches are relevant, and first data of studies combining different combinatorial strategies are discussed below. 

At the European Society for Medical Oncology (ESMO) congress 2019, interim results of the CheckMate-358 phase I/II trial (NCT02488759) investigating the combination of nivolumab and ipilimumab in patients with recurrent or metastatic cervical cancer were presented [186]. Two different dosing combinations were administrated: nivolumab 3 mg/kg every two weeks and ipilimumab 1 mg/kg every 6 weeks (Combo A; *n* = 45), or nivolumab 1 mg/kg and ipilimumab 3 mg/kg every 3 weeks for four doses followed by nivolumab 240 mg every 2 weeks (Combo B; *n* = 46), for ≤24 months until progression or unacceptable toxicity. The objective response rate was higher in patients without prior systemic therapies: 32% (6/19) for combo A and 46% (11/24) for combo B. In patients with prior systemic therapy, the objective response rate was 23% (6/26) for combo A and 36% (8/22) for combo B. Clinical benefit was observed regardless of PD-L1 status. Grade 3–4 AEs were observed in 29% of patients in combo A and in 37% in combo B. 

At ESMO congress 2020, preliminary results of the phase II trial (NCT03495882) investigating balstilimab (anti-PD-1) in combination with zalifrelimab (anti-CTLA-4) in recurrent or metastatic cervical cancer patients were presented [187]. Patients (*n* = 143) received balstilimab 3 mg/kg every 2 weeks in combination with zalifrelimab 1 mg/kg every 6 weeks for up to 2 years. All patients received platinum-based chemotherapy as previous treatment. ORR was 22% (6% CR; 16% PR). Responses were mostly common in the PD-L1+ and SCC patients, but responses were also observed in PD-L1– and AC patients. Treatment was well tolerated, with severe AEs (grade 3+) in 10.5% of the patients. 

#### 4.3.5. Immune Checkpoint Inhibitors and Rare HPV-Related Malignancies (Vulva and Penis)

Based on the FDA approval of pembrolizumab as second-line therapy for PD-L1+ cervical cancer and microsatellite instability-high (MSI-H)/mismatch repair-deficient (dMMR) solid tumors, pembrolizumab is included in the National Comprehensive Cancer Network (NCCN) guidelines for vulvar cancer. The NCCN recommends pembrolizumab for the treatment of recurrent and metastatic vulvar cancers that are PD-L1+ or MSI-H/dMMR. The NCCN guidelines for penile cancer also recommend pembrolizumab as second-line therapy for PSCC patients with recurrent or metastatic disease that is unresectable, MSI-H, or dMMR. 

Several case reports and case series from basket trials have reported on the efficacy of pembrolizumab in vulvar and penile cancer. An impressive response to pembrolizumab after two cycles of immunotherapy was observed in a recurrent vulvar cancer case characterized by PD-L1 and PD-1 mutation [188]. In the basket KEYNOTE-028 trial (NCT02054806), within the advanced PD-L1+ VSCC cohort (*n* = 18), only one (1/18) patient achieved a partial response and seven (7/18) had stable disease after pembrolizumab monotherapy [189]. Preliminary results of the subsequent phase II KEYNOTE-158 (NCT02628067) were presented at the 2021 Society of Gynecological Oncology 52nd Annual Meeting [190]. A total of 101 patients received pembrolizumab 200 mg every 3 weeks until disease progression, unacceptable toxicity, or completion of 35 treatment cycles. ORR was 10.9% (11/101; 1 CR, 10 PR). Responses were durable with a median duration of 20.4 months and occurred in both PD-L1+ and PD-L1– patients. It is unclear whether patients with PD-L1– tumors had MSI-H/dMMR status or tumors with mutational burdens. In another phase II basket trial (NCT02721732), patients with rare tumors that were unresectable or metastatic received pembrolizumab 200 mg every 3 weeks for up to 24 months in the absence of disease progression or toxicity [191]. One case with recurrent PD-L1+ VSCC had a 30% reduction in target lesions after five cycles of pembrolizumab, but discontinued due to grade 3 treatment-related oral mucositis [192]. In the same basket trial, three patients with recurrent, locally advanced, or metastatic PSCC who had progressed on platinum-based chemotherapy triplet were treated with pembrolizumab. One patient with a high-mutational-burden tumor exhibited a partial response, and the other two progressed within 3 months after start of therapy [193]. Moreover, in penile cancer, an additional case series reported durable complete and partial responses in two patients with chemorefractory metastatic disease [194]. The first patient had high tumor mutational burden and a complete response on pembrolizumab that lasted for an impressive 38 months. The second patient had positive PD-L1 expression and a partial response to pembrolizumab that lasted for 18 months. 

Other immune checkpoint inhibitors and checkpoint inhibitor combinations are also being studied. The checkmate-358 phase I/II trial (NCT02488759) investigating nivolumab monotherapy reported a partial responder (HPV-negative VSCC) within the vulvar/vaginal cancer cohort (*n* = 5) [179]. In penile cancer, a partial response to nivolumab and significant tumor shrinkage (<80% reduction in tumor volume) in a patient with HPV-negative, p16-negative advanced chemoradiation refractory cancer was described in a case report [195]. The pretreatment tumor material presented positive expression of PD-L1 on ≥5% of tumor cells, high mutational burden, and MSI absence. Furthermore, a recent phase II basket trial (NCT03333616) examining the combination of nivolumab and ipilimumab enrolled 56 patients, including 6 with advanced penile cancer; unfortunately, no objective response was observed for any of the patients with penile carcinoma [196]. Ongoing exploratory research is studying the predictive potential of tumor mutational burden, PD-L1 status, and other markers to further delineate which patients have the most benefit from the combination of nivolumab and ipilimumab. This might give a better idea as to why such poor responses were observed in patients with penile cancer. 

Altogether, evidence for the use of checkpoint inhibitors in vulvar or penile cancer is based on data that are often extrapolated from case reports and basket trials with very small numbers of patients, and should be interpreted with caution. Well-designed, multicentric clinical trials should be conducted at referral institutions with high caseloads and extensive experience in disease management [197]. 

#### 4.3.6. Predictors of Response to PD-1/PD-L1 Inhibitors

Currently, a combined positive score (CPS) of ≥1 is used for selecting cervical cancer patients to receive pembrolizumab treatment [174,175,177]. The CPS is the combined score for PD-L1 expression in tumor and tumor-field infiltrating immune cells, as calculated by the number of PD-L1-positive cells divided by the total number of tumor cells, multiplied by 100. The CPS positivity is determined by an FDA-approved PD-L1 IHC 22C3 pharmDx test. However, not all patients with CPS positivity show a response to anti-PD-(L)1 therapy, and patients with PD-L1-negative tumors can also benefit from anti-PD-1 and anti PD-L1 therapies. These discordant results in advanced tumors might be due to the observed intrapatient variability in PD-L1 expression. Discordant IHC scores for PD-L1 on tumor cells were observed between primary tumor cells and metastatic tumor cells and within cores of the primary tumor from the same patient [126,198]. Interestingly, with RNA in situ hybridization (RNAish), PD-L1 heterogeneity between core biopsies of the same patient was observed in only 11% instead of 27% of the cases, which highlights the superior consistency of PD-L1 mRNA detection over PD-L1 protein detection [198]. Future research should therefore evaluate whether RNAish could serve as a better biomarker compared to PD-L1 protein expression. 

MSI-H/dMMR status and tumor mutational burden status are approved by the FDA as tumor-site agnostic biomarkers for pembrolizumab [194]. Almost all vulvar and penile cancer patients that responded to pembrolizumab had PD-L1+, MSI-H/dMMR, and/or high-tumoral-mutational-burden tumors. Of note, prospective studies investigating pembrolizumab and other checkpoint inhibitors in vulvar and penile cancer are relatively small. Future studies in bigger cohorts are needed to investigate the role of these biomarkers and explore whether checkpoint inhibitors can be administered in a broader patient population, irrespective of PD-L1, MSI-H/dMMR, or tumor mutational burden status. 

It remains unclear whether differences in treatment outcomes exist between hrHPV− and hrHPV+ cervical, vulvar, and penile cancer patients treated with ICI. What is known is that hrHPV+ and hrHPV− tumors are two molecularly and immunologically distinct subgroups with different levels of immune cell infiltration and immunosuppression, and importantly, improved clinical outcome in HPV+ subgroups [37,84,112,199,200,201]. Future studies should investigate if HPV status affects objective response rate or prognosis in ICI-treated patients. If proven to be true, trials on ICI in HPV-related urogenital cancers should start stratifying patients by HPV status when performing correlative analysis to assess the predictive potential of biomarkers for each subgroup individually. The relation between PD-L1 status, tumor mutational burden, and MSI-H/dMMR status and response to ICI can be masked by the mixture between hrHPV− and hrHPV+ tumors. In addition, beside HPV status, patients with urogenital tumors should be stratified by histology type. In cervical cancer, differences in immune infiltration and PD-L1 expression patterns were observed that are likely to impact responsiveness to PD-1/IC blockade [104,126].

In addition to the above-mentioned biomarkers, CD8+FoxP3+ T-cell rates in cervical tumors and TDLN were found to be associated with PD-1 blockade efficacy in vitro [140], which was confirmed for clinical PD-1 blockade in melanoma using multiplex IHC [202]. In the future, state-of-the-art high-dimensional techniques such as single-cell RNA-Seq (scRNA-Seq), imaging cytometry by time of flight (CyTOF), and spatially resolved RNA transcriptomics should be applied for the discovery of biomarkers beyond PD-L1 expression. In this way, the depth and dimensionality of immune profiling in HPV-related cervical, vulvar, and penile cancer will be greatly improved and may lead to more accurate predictors of clinical response for patients receiving PD-1/PD-L1 inhibitors as well as other novel immunotherapeutic agents. 

### 4.4. Clinical Opportunities for Local Immune Modulation in HPV-Related Cancers

TDLNs are essential players in antitumor immunity and for that reason, serve as potential therapeutic targets for immunotherapy. Indeed, recent studies point to the essential role of TDLN in both CTLA-4 and PD-1 blockade efficacy [203,204,205]. Intratumoral or local ipsilateral administration of ICI ensures optimal access to suboptimally primed tumor-specific T cells in TDLN, in contrast to systemic administration of ICI [203,206], thus facilitating proper priming and differentiation of T-cells. Cancer growth and spread may thus be halted at an early stage, and systemic antitumor T-cell responses could be triggered, which would offer long-lasting protection against recurrences [207,208]. In this way, the tumor acts as its own “vaccine” [209]. 

The lower levels of immune suppression, higher clonal “trunk” neoantigens, and limited tumor heterogeneity in early stage cancers makes is it easier to target and convert the immunosuppressive environment into an immune-stimulating one [210]. Early stage cancers might substantially benefit from a less aggressive approach, such as local administration of low-dose ICI instead of systemic ICI. Monoclonal antibodies accumulate at low levels in off-target tissues when administered locally in contrast to intravenously [211], which minimizes treatment-related AEs. Indeed, preliminary results from the phase I DURVIT trial investigating intratumorally applied durvalumab (anti-PD-L1) in early cervical cancers [212] showed that this strategy is safe and more tolerable than systemically applied anti-PD-L1 therapy; none of the patients had adverse events exceeding grade 2 (Rotman et al.; manuscript in preparation). Moreover, if local administration with low-dose ICI is proven to be not only tolerable but also effective, less extensive surgery needs to be employed in the future, thereby avoiding surgical complications that result from LN dissection (e.g., infection and lymphedema) [213].

Currently, ICI therapy is mainly applied in the recurrent or metastatic setting for HPV-related cancers. Since cervical, vulvar, and penile cancer initially metastasizes to lymph nodes, local modulation of primary tumor and TDLN with low-dose ICI may be of great interest for disease management and reduction of treatment-related morbidity and toxicity. One approach would be to administer local ICI therapy while primary tumor and TDLNs are still in situ. This idea is based on evidence that neoadjuvant immunotherapy can enhance immunity against tumor-specific antigens and eliminate micrometastatic deposits that otherwise would lead to postsurgical relapses [214]. In that manner, this approach would be able to prevent nodal metastatic disease and subsequent invalidating treatments. 

In-depth analysis of the immune microenvironment of TDLN could help discover potential therapeutic targets for local immune modulation. In cervical cancer, T cells in TDLN seem to be less exhausted (i.e., lower levels of multiple immune checkpoints and intermediate rather than high levels of PD-1) compared with T-cells in the primary tumor and may thus be more responsive to ICI [140]. The effects of in vitro PD-1 blockade on HPV 16 E6-specific T cells were indeed more pronounced in cervical cancer TDLN compared to primary tumor. Moreover, high rates of Tregs and suppressive PD-L1 + CD14+ macrophage-like cells were found in TDLNs that contained metastases [98]. These data support the use of local PD-(L)1 blockade for the treatment of cervical cancer, in order to lift loco-regional immune suppression. 

In vulvar cancer, metastatic involvement of TDLN was accompanied by an inflamed microenvironment with immunosuppressive features, marked by hampered activation of migratory DCs, terminal CD8+ effector-memory T-cell differentiation, high regulatory T-cell rates, T-cell activation, and expression of CTLA-4 and PD-1 immune checkpoints [113]. Correlation analyses with primary and metastatic tumor burden suggested respective roles for Tregs and suppression of ICOS+ T helper cells in early metastatic niche formation and for CD14+ lymph-node-resident DCs and terminal T-cell differentiation in later stages of metastatic growth. TDLN-targeting interventions combining CTLA-4 (in earlier stages) and PD-1 blockade (in later stages) should be considered for vulvar cancer patients, in order to reinvigorate memory T cells and prevent metastatic spread and growth. 

In penile cancer, the major finding up to now is the high expression of various checkpoint molecules in an increasing fashion from nonmetastatic lymph nodes, to metastatic lymph nodes, to primary tumors (Rafael et al.; manuscript in preparation). These results support clinical exploration for TDLN-targeted approaches based on immune checkpoint blockade therapy.

## 5. Conclusions and Future Perspectives

Immune cells present in the microenvironment of HPV-related (pre-)malignant lesions of the cervix, vulva, and penis are often functionally impaired and inhibited in their antitumor function. Better understanding of evasion strategies of HPV and HPV-driven tumors, especially as to how the immune escape mechanisms differ per anatomic/tissue site, is of high importance due to the distinct clinical behavior of HPV-related malignancies. Based on current knowledge on evasion strategies of HPV and HPV-driven tumors, specific strategies at different stages of disease development can be employed to restore immune cell function in HPV-related tumor microenvironments. Since immune escape mechanisms present in the tumor microenvironment become more and more complex with progression of the disease, early interventions (such as imiquimoid, therapeutic vaccination, or local immune checkpoint blockade) in the premalignant setting should be considered to overcome early immune escape and progression to full-blown malignancy. Many prospective studies on imiquimod and therapeutic vaccines have shown the clinical benefit of adding such agents to the treatment regimen. In advanced disease settings, PD-1/PD-L1 monotherapy has provided encouraging but modest clinical efficacy with often short-lived benefit. The failure of PD-1/PD-L1 monotherapy may be attributable to a whole plethora of additional immune evasion mechanisms causing primary or adaptive resistance. In order to increase and prolong clinical efficacy in advanced HPV-related cancers, combinations of different immunotherapeutic agents should be explored. Early stage cancers might benefit from less aggressive approaches, such as neoadjuvant and/or local administration of low-dose ICI. Stratification based on histological subtype and HPV status should be taken into consideration in trials investigating immunotherapeutic agents, as distinct immune microenvironments and immune escape mechanisms have been observed between AC and SCC, and hrHPV– and hrHPV+ tumors. Further well-designed, multicentric studies are especially needed to investigate the efficacy of checkpoint inhibitors in vulva and penile cancer and to investigate whether checkpoint inhibitors can be administered in a broader patient population, irrespective of PD-L1+, MSI-H/dMMR, or tumor mutational burden status. 

## Figures and Tables

**Figure 1 jcm-11-01101-f001:**
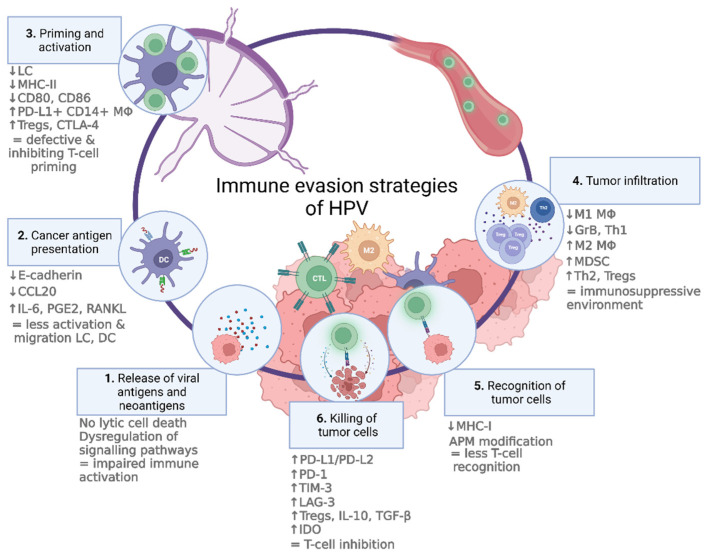
This figure shows an adapted version of the cancer immunity cycle to illustrate the intracellular and extracellular evasion strategies of human papillomavirus (HPV). (1) Release of viral antigens and neoantigens; (2) cancer antigen presentation (chemokine CCL20 (CCL20), interleukin-6 (IL-6), prostaglandin E2 (PGE2), receptor activator of NF-KB ligand (RANKL), Langerhans cells (LCs), and dendritic cells (DCs)); (3) priming and activation (major histocompatibility complex II (MHC-II), programmed death-ligand-1 (PD-L1), regulatory T cells (Tregs), and cytotoxic T-lymphocyte-associated protein (CTLA)); (4) tumor infiltration (pro-inflammatory macrophages (M1 MΦ), granzyme B (GrB), T-helper cells 1 (Th1s), anti-inflammatory macrophages (M2 MΦ), and myeloid-derived suppressor cells (MDSCs)); (5) recognition of tumor cells (antigen-processing machinery (APM) modification); (6) killing of tumor cells (PD-1 receptor (PD-1), T-cell immunoglobulin and mucin-domain-containing-3 (TIM-3), lymphocyte activation gene-3 (LAG-3), transforming growth factor (TGF-β), indoleamine 2,3-dioxygenase (IDO)). Created with BioRender.com (last accessed on 15 February 2022).

**Figure 2 jcm-11-01101-f002:**
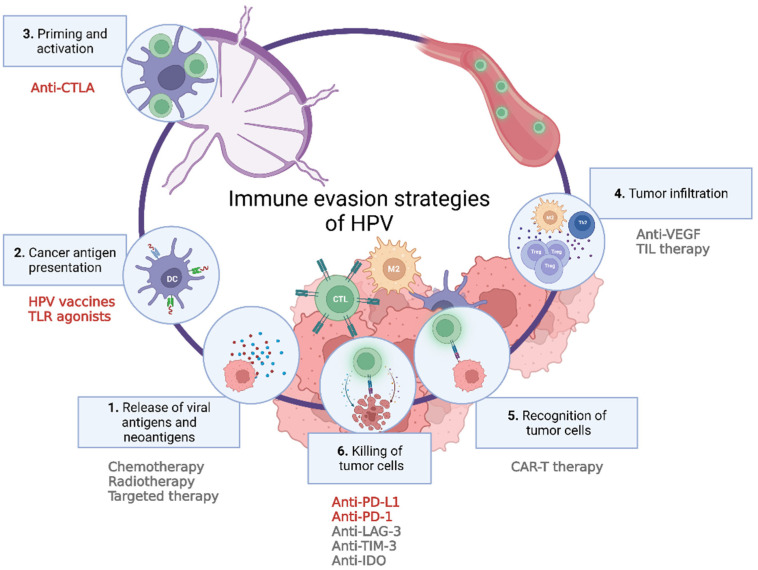
This figure highlights examples of strategies that are currently being employed in preclinical and clinical settings to restore immune function in human papillomavirus (HPV) infected tumor microenvironment. The strategies marked in red are discussed in this review. Toll-like receptor (TLR); cytotoxic T-lymphocyte-associated protein (CTLA); vascular endothelial growth factor (VEGF); tumor-infiltrating lymphocytes (TIL) therapy; chimeric antigen receptor T-cells (CAR-T) therapy; programmed death-ligand-1 (PD-L1); PD-1 receptor (PD-1); T-cell immunoglobulin and mucin-domain-containing-3 (TIM-3); lymphocyte activation gene-3 (LAG-3); indoleamine 2,3-dioxygenase (IDO). Created with BioRender.com (last accessed on 15 February 2022).

**Table 1 jcm-11-01101-t001:** Results of clinical trials investigating ICI as treatment for cervical cancer.

Study	Medication	Population	ORR	N	AEs
ICI—Monotherapy					
KEYNOTE-826 (NCT03635567)—phase III					
	Pembrolizumab (anti-PD-1)	Persistent/ recurrent/ metastatic	PFS: 10.4 months vs. 8.2 months for control arm	617	49% ≥ grade 3 pembro vs. 42% ≥ grade 3 control arm
		CxCa	(as primary outcome)		
GOG 3016/ ENGOT-cx9 (NCT03257267)—phase III					
	Cemiplimab	Recurrent/ metastatic	OS: 12.0 months vs. 8.5 months for control arm	304	NA
	(anti-PD-1)	CxCa	(as primary outcome)		
KEYNOTE-028 (NCT02054806)—Phase Ib					
	Pembrolizumab	PD-L1^+^ advanced	17% (no CR, 4 PR, 3 SD)	24	21% grade 3
		CxCa			
KEYNOTE-158 (NCT02628067)—Phase II					
	Pembrolizumab	Recurrent/ metastatic	12% (3 CR, 9 PR)	98	12% grade 3–4
		CxCa			
CheckMate 358 (NCT02488759)—Phase I/II					
	Nivolumab	Recurrent/ metastatic	26% (3 CR, 1 PR)	19	21% grade 3–4
	(anti-PD-1)	CxCa			
NCT02257528 Phase I/II					
	Nivolumab	Persistent/ recurrent	4% (1 PR)	26	32% grade 3–4
		CxCa			
NCT03104699 Phase II					
	Balstilimab	Recurrent/ metastatic	15% (5 CR, 16 PR)	161	12% ≥ grade 3
	(anti-PD-1)	CxCa			
NCT03104699 Phase II					
	Ipilimumab	Recurrent/ metastatic	3% (1 PR)	42	29% ≥ grade 3
	(anti-CTLA-4)	CxCa			
ICI—combination therapy					
GOG 9929 trial (NCT01711515)—phase I					
	Ipilimumab + CRT	Node-positive CxCa	1-year OS 90%, PFS 81%	32	9.5% grade 3
			(as secondary outcome)		
NCT02383212 Phase I					
	Cemiplimab	Persistent/ Recurrent	10% both cohorts (1 PR)	10	10% ≥ grade 3 mono
	Mono or combo with hfRT	CxCa			10% ≥ grade 3 combo
Checkmate-358 (NCT02488759)—Phase I/II					
	Combo A: nivo + ipi	Recurrent/ metastatic	A: 32% w/o PST 23% with PST	A: 45	A: 29% grade 3–4
	Combo B: nivo + ipi, followed by nivo	CxCa	B: 46% w/o PST 36% with PST	B: 46	B: 37% grade 3–4
NCT03104699 Phase II					
	Bastilimab (anti-PD-1) + Zalifrelimab (anti-CTLA-4)	Recurrent/ metastatic	22% (8 CR, 23 PR).	143	10.5% ≥ grade 3
		CxCa			

Abbreviations: N, number of patients; AEs, adverse events; CxCa, cervical cancer; ICIs, immune checkpoint inhibitors; CR, complete response; PR, partial response; SD, stable disease; hfRT, hyperfractionated radiotherapy; mono, monotherapy; combo, combination therapy; nivo, nivolumab; ipi, ipilimumab; pembro, pembrolizumab; w/o, without; PST, prior systemic therapies; CRT, chemoradiation therapy; OS, overall survival; PFS, progression-free survival; NA, not applicable.

## Data Availability

Not applicable.

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
