# Peer review of "Immunotherapeutic Approaches for the Treatment of HPV-Associated (Pre-)Cancer of the Cervix, Vulva and Penis"

_jcm, 2022, doi:10.3390/jcm11041101_

Round 1
Reviewer 1 Report
This paper extensively describes in details the clinical spectrum and nature of HPV-associated (pre-)cancers, focusing the immunotherapeutic approaches. Since the cervical, vulvar and penile cancers and their treatment are present in many the practice of several disciplines, affect both females and males, this review of the present status of the immunetherapy strategies might be of interest of many practicioners. Since the description of the cellular events of the HPV immune evasion strategies is very complex in the chapter 3, as well as the restoration of the immune cell function in chapter 4, a few simply figures showing the evidenced steps and a list of the abbreviations would help the better understanding. Further, shortening the most detailed cellular descriptions may help the reception and reduce the time for reading.
This is an ambitious and well written state-of-the-art review paper of the chosen topic, and considering the high standard of it, despite its extended lengths and depths.
Author Response
Dear Reviewer,
We would like to thank you for the willingness to consider our review and the helpful and constructive feedback. We have included a list of abbreviations at the end of the review. We also added two figures to the review. The first figure (Section 3. Immune evasion strategies of HPV and HPV-related cancer and precursor lesions of the cervix, vulva and penis) summarizes the most important immune evasion strategies of HPV and the second figure (Section 4. Restoring immune cell function in the HPV-related tumor microenvironment) highlights different strategies to restore immune functions in HPV-infected tumor microenvironment. Abbreviations used in the figures are also listed in the figure legends. We believe this suggestion has indeed improved the readability of the review. We also would like to point out that we added such detailed cellular descriptions in case readers are specifically interested in those details, and hope Figure 1 helps those who are not interested in such details.
Many thanks for taking the time to review our article.
Sincerely,
Tynisha Rafael
Reviewer 2 Report
This paper represents very interesting and detailed review of immunotherapeutic approaches for the treatment of HPV-associated (pre-)cancer of the cervix, vulva and penis. The manuscript is excellently written, easy to read and to understand. This reviewer has only a few minor suggestions:
Page 6, line 292 – please, add a full stop after the references “[98,111]”
Page 8, line 381– please, add a comma before “but”
Page 8, line 385 – “in-vitro” or “in vitro” Please, check the whole manuscript and standardize.
Page 8, line 387, Page 13, line 563 and Page 14, line 614 – please, add a comma before “but”
Page 13, line 577 – please, add a comma after “therapy”
Page 15, line 682 – please, add a comma after “cervical cancer”
Author Response
Dear Reviewer,
Thank you for taking the time to review our article and the provided comments on punctuations. We have edited the review accordingly.
Sincerely,
Tynisha Rafael
Reviewer 3 Report
The paper presents HPV in etiology in cancer of the cervix, vulva and penis. Vaccination against HPV reduces pre-malignant lesions and cancer related to persistent HPV infection. Immune evasion strategies of HPV and HPV-related cancers play a major role in cancer development. Standard therapies (surgery, chemo-radio therapy) have poor results, immunotherapy can be an additional option to improve the effect of treatment.
The authors highlight the importance of early interventions as early as possible in the pre-malignant stage to halt immune escape mechanisms and prevent cancer development. Checkpoint inhibitors blocking PD-1, PD-L1 and/or CTLA-4, according to the presented clinical trials seem promising yet further studies in a broader population are necessary .
The authors give a full and broad review with sufficient background information, grounding the paper well. They explore immune evasion strategies in depth, and describe methods of restoring immune cell function in the HPV-related tumor microenvironment. Data from clinical trials support such a line of treatment in the cancers this paper concerns.
There is a large number of references despite the majority being older than five years, the paper cites recent work.
Author Response
Dear Reviewer,
Many thanks for taking the time to review our article and the provided comments.
Sincerely,
Tynisha Rafael